# Impacts of Astaxanthin Supplementation on Walking Capacity by Reducing Oxidative Stress in Nursing Home Residents

**DOI:** 10.3390/ijerph192013492

**Published:** 2022-10-18

**Authors:** Ryosuke Nakanishi, Miho Kanazashi, Masayuki Tanaka, Minoru Tanaka, Hidemi Fujino

**Affiliations:** 1Department of Rehabilitation Science, Kobe University Graduate School of Health Sciences, Kobe 654-0142, Japan; 2Department of Physical Therapy, Kobe International University, Kobe 658-0032, Japan; 3Department of Physical Therapy, Prefectural University of Hiroshima, Hiroshima 723-0053, Japan; 4Department of Physical Therapy, Okayama Healthcare Professional University, Okayama 700-0913, Japan; 5Department of Rehabilitation Science, Osaka Health Science University, Osaka 530-0043, Japan

**Keywords:** endurance capacity, oxidative stress, astaxanthin, older adults, muscle strength

## Abstract

Oxidative stress is associated with deterioration of endurance and muscle strength, which are mostly accompanied by aging. Astaxanthin supplement has excellent antioxidant activity without any pro-oxidative properties. In this study, we investigated how astaxanthin supplementation affects walking endurance and muscle strength in nursing home residents. Healthy elderly individuals (age: 67 to 94) were divided into two groups: 13 subjects received a daily dose of 24 mg of astaxanthin for 16 weeks (astaxanthin group) and 11 subjects received a placebo (placebo group). These subjects were compared using body component measurements, serum d-ROM levels, the distance of 6-min walking, blood lactate levels after the 6-min walking test, and muscle strength. After supplementation, the levels of d-ROMs and blood lactate after the 6-min walking test in the astaxanthin group significantly decreased compared with the placebo group (*p* < 0.05). Additionally, the walking distance was significantly higher in the astaxanthin group than in the placebo group (*p* < 0.05), despite a significant reduction in lactate levels after 6-MWT (*p* < 0.05). However, no significant intergroup differences were observed in muscle mass and strength. Astaxanthin supplement for 16 weeks is effective to increase the endurance capacity of the elderly. Astaxanthin supplement suppresses d-ROMs at rest and lactic acid production after the 6-min walk test. In contrast, astaxanthin supplement did not show significant intergroup differences in the muscle mass and strength. Therefore, the effect was most likely accompanied by an increase in endurance instead of an increase in muscle strength.

## 1. Introduction

One of the variables that contribute to the age-related decline in endurance is oxidative stress, which strongly disrupts energy metabolism [1]. Oxidative stress also decreases muscle strength in older adults [2,3]. As a result, age-related oxidative stress conditions may decrease endurance by interfering with energy metabolism and muscular strength.

Continuous low-intensity exercise is well known to reduce oxidative stress [4] and increase endurance in older adults [5,6]. However, older adults are at increased risk for cardiac events and musculoskeletal injuries brought on by exercise. A previous study revealed that almost all older adults were unable to continue their exercise because of constraints, such as health problems, pain, and physical environment [7]. Hence, it is important to develop a new method that may be used to replace exercise for any older adults and be a sustainable countermeasure for an increase in oxidative stress that results in the loss of endurance capacity.

Antioxidant supplements are beneficial for preventing age-related oxidative stress and tissue aging, including immunity of T cells, vascular smooth muscle cells, endothelial cells, and mitochondrial dysfunction in skeletal muscle [8].

Astaxanthin supplement is a red carotenoid that provides various benefits to prevent cancer by anti-proliferation, preventing muscle atrophy by anti-apoptosis, and anti-inflammation [9,10,11]. Furthermore, astaxanthin supplement has excellent antioxidant activity that quenches singlet oxygen and inhibits lipid peroxidation, and it is effective as an anti-oxidant without processing any pro-oxidative properties [11,12]. Thus, astaxanthin is likely the ideal supplement for elderly individuals who have a reduced capacity to relieve oxidative stress, a risk, and a low motivation. In fact, the benefits of astaxanthin supplementation on endurance in the elderly have been reported. According to Liu et al., healthy elderly adults who are capable of performing activities of daily living in their homes without assistance can increase their endurance, muscle strength, and muscle size by taking astaxanthin supplements together with endurance exercises [13]. They also indicated improved muscle endurance following an increase in serum level of astaxanthin concentration in another report [14]. However, it is not possible to say whether astaxanthin supplements alone or astaxanthin combined with exercise has an impact on these factors in these studies. In addition, participants in the previous study were community-dwelling elders, and the components of their daily diet are different. Thus, the effectiveness of astaxanthin supplements taken alone is unknown due to a lack of dietary information important for reducing oxidative stress. Therefore, the goal of this study was to determine whether taking an astaxanthin supplement without exercising can improve walking endurance in elders living in nursing homes who followed the same diet after oxidative stress has reduced. This is the first study to show how astaxanthin supplements affect endurance in the elderly whose diets are well monitored by regulating oxidative stress levels. The unique aspect of astaxanthin supplementation is that the subjects could improve endurance without mechanical stress to the body. If this study is validated, a new preventative strategy for losing endurance can be established for older adults.

## 2. Materials and Methods

### 2.1. Subjects

Thirty healthy elderly adults living in nursing homes volunteered to participate in this study and were randomly assigned to the placebo group (n = 15) or the astaxanthin group (n = 15). Six subjects dropped out after randomization: five for medical reasons unrelated to the treatment and one for personal reasons. Twenty-four subjects (n = 11, n = 13) completed the study, and these allocations are shown in Figure 1. The subjects received sufficient instructions in how to use supplements properly. They were instructed to avoid any intense or unaccustomed strength training or endurance training throughout the study period and complied. They were also asked to refrain from strenuous exercise for 24 h before the measurement sessions and not to consume caffeinated drinks on the day of testing. This study was conducted following the guidelines outlined in the Declaration of Helsinki. All procedures involving human subjects were approved by the Ethics Committee on Human Research at Kobe University (No. 277-1). Written informed consent was obtained from all subjects.

### 2.2. Inclusion and Exclusion Criteria

We included healthy older adults who resided in a nursing home and consumed a set meal prepared there. However, we excluded the following respondents: (1) subjects who cannot take supplements due to cognitive decline, (2) subjects who cannot walk without assistance, and (3) subjects who are involved in regular training or active strength training programs.

### 2.3. Experimental Protocol

Before taking the supplement, each subject underwent the body component measurement, d-ROMs test, 6-min walking test (6 MWT), and muscle strength test for determination of “before data”. Each subject started taking supplements the day after the measurement. For 16 weeks, the supplement was taken twice daily with breakfast and dinner. After the experiment period, these items were measured as “after data”. All measurements were conducted in an environmentally controlled room at 25 ± 2 °C.

### 2.4. Supplementation

Two groups were obtained using a double-blind placebo-controlled design with randomization (placebo group and astaxanthin group). Group randomization assignments were generated by an unblinded statistician. Additionally, placebo capsules and astaxanthin capsules were generated to look identical by the manufacturer. The subjects in the astaxanthin group received supplementation with astaxanthin (AstaReal act; astaxanthin, 24 mg/day; tocotrienol, 40 mg/day; ascorbic acid, 30 mg/day), whereas those in the placebo group received supplementation without astaxanthin (tocotrienol, 40 mg/day; ascorbic acid, 30 mg/day). In a safety review of a human study, it was determined that supplementing with 24 mg/kg/day of astaxanthin was not harmful [15]. It is therefore demonstrated in this study that taking this amount of astaxanthin is safe.

### 2.5. Assessment of Body Weight, Percent Muscle Mass, and Percent Body Fat

Measurements of body weight, percent muscle mass, and percent body fat were performed using bioelectrical impedance analysis (DF 860, Yamato Co., Tokyo, Japan). The measurements were taken prior to all measurements and performed in the standing position.

### 2.6. The d-ROMs Test

Before the muscle strength test and 6 MWT, serum samples were taken for the d-ROMs. Blood samples (~100 μL) were drawn from the fingertips 12 h after the last dose of each supplement. The collected blood sample was centrifuged at 13,000× *g* for 2 min at 37 °C, and the supernatant was collected. The supernatant samples reacted to the d-ROMs kit (Diacron International, Grosseto, Italy) and were detected using a spectrophotometric machine (FREE CARRIO DUO, Diacron International) using absorption at 505 nm. The results of the d-ROM level are expressed in an arbitrary unit called the Carratelli unit (U. CARR) [16]. The normal values of the test are between 250 and 300 U. CARR., where 1 U. CARR. corresponds to 0.8 mg/L H_2_O_2_.

### 2.7. The Muscle Strength Test

As the measure of maximum muscle strength, the peak torque obtained during maximum voluntary isometric contraction of both sides of knee extension and handgrip was taken two times. The peak torque of knee extension and handgrip strength was measured by using knee extension isometric contraction measurement devices (GT-30, OG Giken, Okayama, Japan) and handgrip dynamometer (TKK5401, Takei Scientific Instruments Co., Ltd., Niigata, Japan).

### 2.8. The 6-min Walking Test and Lactate Acid Change

The 6 MWT was conducted according to a standardized protocol [17] using an internal hallway with the 50 m distance marked by colored tape on the floor. Subjects’ walking distances were measured at the end of a 6-min walk. In addition, before the walk started and at the end of the 6-min walk, blood lactate of the subjects was measured through the fingertips using a lactate sensor (Lactate Pro2, ARKRAY, Inc., Tokyo, Japan). The lactic acid level showed the rate of change from prior supplementation.

### 2.9. Statistical Analysis

Data are reported as median (min–max). All statistical analyses were performed using PRISM software version 8.0 (Intuitive Software for Science, San Diego, CA). *p* values less than 0.05 were considered statistically significant. Body components, d-ROMs, the walking distance during 6 MWT, and muscle strength were all analyzed using two-way repeated measure ANOVA (supplements; placebo and astaxanthin) × (time; before and after). The Bonferroni post hoc test was utilized to determine any changes within the group from the baseline to the intervention when a group diet interaction was discovered. Changes in the levels of lactic acid were analyzed by an unpaired sampled *t*-test. The sample size required for a two-way repeated measures ANOVA for this study was calculated using the G*power 3.1 software (Heinrich Heine University, Dusseldorf, Germany), and more than twenty subjects were required for this study (effect size, 0.35; α error, 0.05; and power, 0.80). These values are derived from a previous study on the effects of supplementation for 16 weeks on the oxidative stress biomarker (protein carbonyls) in older adults [18].

## 3. Results

### 3.1. The Body Weight, Percent Muscle Mass, and Percent Body Fat

Table 1 summarizes pre- and post-intervention for body composition supplementation. The characteristics of twenty-three volunteers who completed the supplement intervention were recorded. The two groups were well matched for age and body composition. No significant differences were found in the body composition in any data.

### 3.2. The Value of d-ROMs at Rest

Figure 2 shows the changes in d-ROM values before and after the supplement intervention. Prior to the placebo supplement, the value of d-ROMs was 364 (273–501) U.CARR, and after the placebo supplementation, it was 366 (302–434)  U.CARR. The value of d-ROMs before taking astaxanthin supplement was 331 (215–417) U.CARR, and after taking astaxanthin supplement, it was 292 (202–384)  U.CARR. A significant two-way interaction was found (time × supplement, *p* < 0.05, F = 5.17). In addition, as a result of post hoc testing, the value of the d-ROMs after the astaxanthin supplement was lower than it was before the astaxanthin supplement.

### 3.3. The Walking Distance of the 6-min Walking Test

Figure 3 shows the changes in the value of the walking distance before and after supplement intervention. The values of the walking distance before and after the placebo supplement were 326 (68–438) m and 314 (56–448) m, respectively. Walking distance values before and after the astaxanthin supplement were 333 (107–478) m and 371 (180–494) m, respectively. There was a significant two-way interaction (time × supplement, *p* < 0.05, F = 6.12). In addition, as a result of post hoc testing, the value of the walking distance after the astaxanthin supplement was lower than it was before the astaxanthin supplement.

### 3.4. The Value of Blood Lactate after 6-min Walking Test

Figure 4 shows the percentage of blood lactate levels at pre- and post-6 MWT before taking the supplement. The percentages of blood lactate levels prior to taking the supplement at pre-6 MWT were 100.0% (60.0–370.0) and 122.2% (64.3–280). The percentages of blood lactate levels prior to taking the supplement at post-6 MWT were 116.7% (63.6–237.5) and 90.0% (21.9–135.7). There was a significant difference in the change of rate of blood lactate levels prior to taking the supplement at post-6 MWT.

### 3.5. The Value of Muscle Strength

The changes in the value of muscle strength before and after supplement intervention are shown in Figure 5. The value of knee extensor muscle strength before placebo supplement was 16.5 (5.0–31.0) kg, and after placebo supplement it was 14.7 (4.0–28.3) kg. The values of knee extensor muscle strength before and after astaxanthin supplement were 15.0 (5.0–33.0) kg and 12.8 (6.7–27.7), respectively. The values of handgrip strength before and after placebo supplement were 19.6 (7.3–33.0) kg and 20.0 (10.3–26.5) kg, respectively. The values of handgrip strength before and after astaxanthin supplement were 18.6 (6.8–32.2) kg and 17.5 (6.5–29.7) kg, respectively. There were no significant differences in the knee extensor muscle strength or handgrip strength in any data.

## 4. Discussion

The main finding of the current study was that taking an astaxanthin supplement for 16 weeks increased walking endurance while also reducing oxidative stress in the elderly living in a nursing home. The effect was most likely accompanied by an increase in energy metabolism instead of an increase in muscle strength. Oxidative stress is strongly affected by the daily diet component, i.e., eating a diet high in lipids causes more oxidative stress, whereas eating a diet high in vegetables and fruits reduces oxidative stress [19]. Participants in this study live in the nursing home where they were provided almost the same dietary components every day. Therefore, it was possible to measure only the effects of the supplement, ignoring the effects of the daily diet, and astaxanthin supplementation is most likely an effective intervention to reduce oxidative stress in elderly adults.

The value of d-ROMs is a well-known indicator of oxidative stress marker and indicates a high oxidative condition when it exceeds 300 U.CARR [20]. Thus, our findings show that almost all the participants had oxidative stress before taking supplements. In addition, the walking distance of the 6 MWT was lower than the predictive equations of healthy adults according to age, height, and body weight: (men: 7.57 × height (cm)) − (5.02 × age) − (1.76 × weight (kg)) − 309 m) (women: (2.11 × height (cm)) − (2.29 × age) − (5.78 × weight (kg)) + 667 m) [21]. Likewise, handgrip strength and knee extensor strength were below the age-related average in the previous study [22,23,24]. According to Sastre et al., oxidative stress is one of the factors contributing to age-related decrease in endurance which strongly disrupts energy metabolism [1]. In addition, oxidative stress reduces muscle strength, which affects the endurance of older adults [2,3]. Thus, age-related oxidative stress may have an impact on a reduction in endurance by disrupting the energy metabolism and muscular strength.

Taking astaxanthin supplement decreased the value of d-ROMs during rest. Moreover, taking astaxanthin supplements increases the walking distance and decreases the value of lactic acid produced during the 6 MWT without increasing muscle strength. Therefore, in this study, it is most likely that taking astaxanthin supplements increased endurance due to changes in energy metabolism by reducing oxidative stress. Oxidative stress is classically defined as a serious imbalance between prooxidants such as reactive oxygen species (ROS) production and antioxidant defenses [25,26]. Astaxanthin supplement is a red carotenoid that removes these radicals by disrupting free radical chain reactions or reacting with them to produce harmless products [25,27]. In addition, Lawlor et al. [28] indicated that taking an astaxanthin supplement increases the activity of various antioxidant enzymes such as superoxide dismutase and catalase. Thus, it is possible that taking astaxanthin supplement not only inhibited ROS production but also increased antioxidant enzyme activity, resulting in decreased d-ROMs.

Oxidative stress is closely associated with mitochondrial function, and when the mitochondrial function is insufficient during energy metabolism, it will decrease ATP production and increase lactic acid accumulation, which easily induces fatigue during activities [29,30]. In addition, oxidative stress may disrupt capillary density, the blood supply to adjacent muscle fibers, and the ability to eliminate lactic acid, resulting in a decrease in endurance capacity [31,32,33,34]. However, in this study, taking an astaxanthin supplement reduced the level of lactic acid produced after walking, which may have increased endurance due to change of energy metabolisms.

In fact, a previous study found that astaxanthin supplementation reduces age-related oxidative stress in dogs, which in turn prevents mitochondrial dysfunction [35,36]. The authors also indicated that astaxanthin supplementation increases ATP production following an increase in respiratory chain complex activity in the mitochondria [35]. Moreover, astaxanthin supplementation can change capillaries by regulating oxidative stress under atrophied muscle [37]. Therefore, it is hypothesized that astaxanthin supplements not only activate ATP production but also decrease lactic acid production during exercise by increasing mitochondrial function and the capillary number. Other mechanisms have also been proposed for decreasing lactic acid production. Astaxanthin supplements increase accelerated lipid utilization during energy metabolism and decrease lactic acid during exercise [38]. When energy production during exercise depends on the supply from carbohydrates, not lipids, the intramuscular cells will increase lactic acid accumulation. Altogether, astaxanthin may alter the supplied substrate and have an impact on energy metabolism. For these reasons, suppressing oxidative stress with long-term astaxanthin supplementation increases walking endurance capacity in older adults by reducing lactic acid production.

A limitation of this study is that we did not determine the mitochondrial function, the density of capillary in the muscle, and supplied substrates rate. Future research, however, should be conducted to determine the potential effects of astaxanthin supplementation individually on older adults. Another limitation is that we only studied the effects of the same amount of astaxanthin supplement; however, we did not attempt to elucidate the effects of different weights of astaxanthin. We used soft capsules for our experiments to hide the appearance of astaxanthin color. Therefore, all subjects received the same amount of astaxanthin supplement owing to the difficulty of changing the content of each capsule. Future studies should determine the effects of the amount graded by weight on these results. In this study, a height measurement error was observed. Correcting the alignment is necessary to measure the height of the elderly; furthermore, it is speculated that this measurement error may have occurred as a result of the degree of correction. Minimizing measurement errors in elderly research is of paramount importance in spite of performing alignment corrections within a range that does not cause discomfort to the subjects.

## 5. Conclusions

The current study demonstrated that supplementation of astaxanthin for 16 weeks effectively increased the endurance capacity of older adults by suppressing d-ROMs at rest and by reducing lactic acid production during the 6-min walking test. However, there were no significant intergroup differences with regard to muscle mass and strength. Therefore, the effect was most likely accompanied by an increase in endurance instead of an increase in muscle strength.

## Figures and Tables

**Figure 1 ijerph-19-13492-f001:**
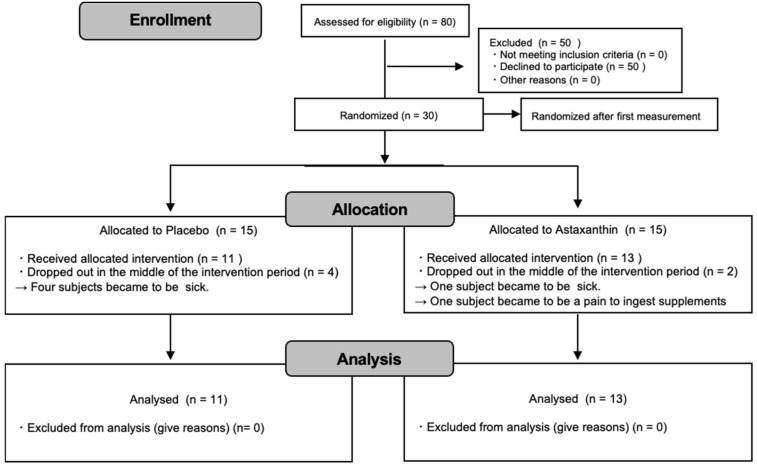
A flowchart showing the distribution of participants the trial.

**Figure 2 ijerph-19-13492-f002:**
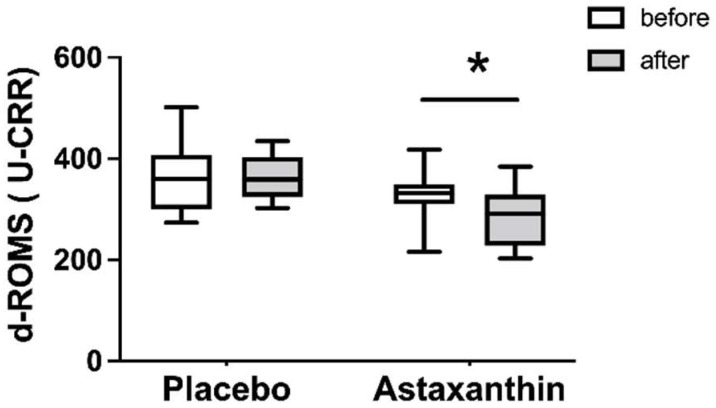
The values of d-ROMs before and after taking the supplement are represented in white and gray bar, respectively. The data are presented as median (min–max). * significantly different compared with before group of each intervention, *p* < 0.05.

**Figure 3 ijerph-19-13492-f003:**
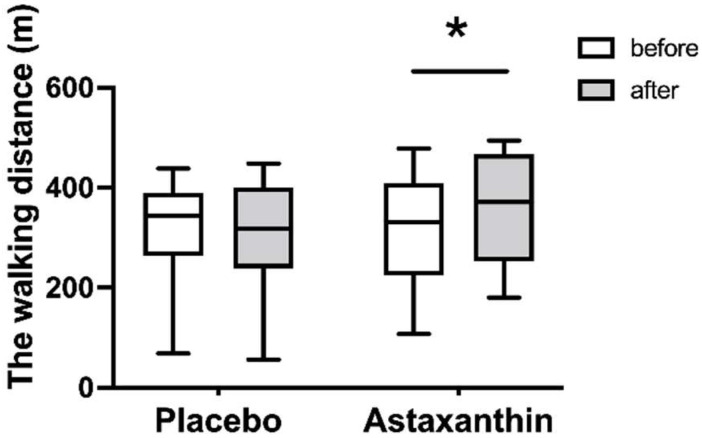
The walking distances of 6-min waking test before and after taking the supplement are represented in white and gray bar, respectively. The data were presented as median (min–max). * significantly different compared with prior to taking astaxanthin, *p* < 0.05.

**Figure 4 ijerph-19-13492-f004:**
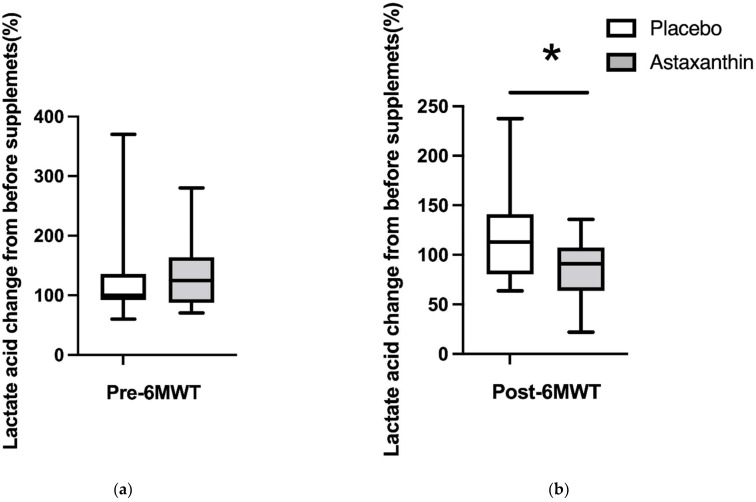
The percentage of blood lactate levels before ingesting the supplement at pre 6 MWT (**a**) and post 6 MWT (**b**). The value of the blood lactic acid placebo ingesting is represented by the white bar and astaxanthin is the grey bar. The data are presented as median (min–max). * significantly different compared with before 6 MWT, *p* < 0.05.

**Figure 5 ijerph-19-13492-f005:**
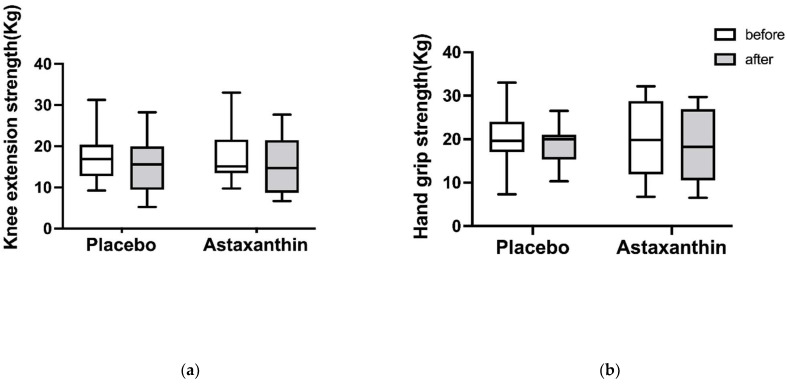
The values of knee extension strength (**a**) and hand grip strength (**b**) before and after taking the supplement are represented by the white and gray bar, respectively. The data are presented as median (min–max).

**Table 1 ijerph-19-13492-t001:** The body weight, percent muscle mass, and percent body fat.

	Placebo Group		Astaxanthin Group	
Median (Min–Max)	Before	After	Before	After
Age (years)	84 (68–94)	84 (68–95)	84 (67–93)	84 (67–93)
Height (cm)	147.0 (132.0–168.0)	146.0 (133.0–170.0)	153.3 (145.5–167.0)	153.5 (143.0–167.0)
Body weight (kg)	48.6 (32.2–66.2)	48.1 (32.5–66.3)	53.0 (33.8–78.7)	51.8 (35.2–78.6)
BMI (kg/m^2^)	23.5 (17.2–27.3)	23.9 (16.5–28.3)	23.7 (14.5–32.9)	23.7 (15.0–33.1)
Waist circumference (cm)	89.0 (68.0–94.0)	89.0 (68.0–95.0)	86.5 (65.0–101.0)	83.5 (68.0–105.0)
Percent muscle mass (%)	24.8 (19.9–34.5)	23.8 (19.9–30.2)	24.2 (14.5–32.9)	24.5 (19.9–30.3)
Percent body fat (%)	32.3 (10.6–45.4)	35.6 (9.4–46.7)	31.1 (23.8–49.0)	34.0 (25.1–50.0)

The data were presented as median (min–max).

## Data Availability

The data presented in this study are available on request from the corresponding authors.

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
