# Peer review of "Impacts of Astaxanthin Supplementation on Walking Capacity by Reducing Oxidative Stress in Nursing Home Residents"

_ijerph, 2022, doi:10.3390/ijerph192013492_

Round 1

Reviewer 1 Report

The main aim of the paper: „Impacts of walking capacity on astaxanthin supplementation via decreasing oxidative stress in nursing home residents“ was to determine whether taking an astaxanthin supplement without exercising can improve walking endurance in elders living in nursing homes who derived the same diet after oxidative stress has reduced.

I would like to appretiate the efforts of the authors, the topic is interesting, but there are some points, which need to be explained, improved or changed.

Did all members of the experimental group receive the same amount of astaxanthin supplement or was the amount graded by weight?

Table 1 shows the change in body height before and after the intervention. Why did the height of the study participants change? Do you have an explanation for this? The change occurred in both directions…

I recommend to expand the conclusion a bit with the practical application of the results.

Participants were instructed not to engage in endurance activities. Was compliance with this recommendation verified?

Author Response

Response to the comments of Referee #1,

We thank the Referee for carefully reading our manuscript and for the helpful comments.

Comments:

#1. Did all members of the experimental group receive the same amount of astaxanthin supplement or was the amount graded by weight?

Response: We are aware that astaxanthin ingredients may have different effects by the subject's body weight. But we used soft capsules to hide the appearance of the color of the supplement in the present study. It was difficult to change the content of each capsule. Therefore, all subjects received the same amount of astaxanthin supplement in soft capsules (6mg astaxanthin/capsule).

#2. Table 1 shows the change in body height before and after the intervention. Why did the height of the study participants change? Do you have an explanation for this? The change occurred in both directions…

Response: We do not have a definitive answer on the discrepancy measurement error of the body height between before and after supplementation. However, we suspect that measurement error is the following of these causes.

  1. Subjects are elderly and have poor alignment of the whole body.
  2. We performed alignment correction within a range that did not cause discomfort to the subjects.

From these things, I guess that a measurement error occurred when correcting the alignment.

#3. I recommend to expand the conclusion a bit with the practical application of the results.

Response: We have corrected the conclusion according to the referee’s advice.

Pages 9, lines 309-314, from “The current study demonstrates that suppressing oxidative stress by taking astaxanthin supplement for 16 weeks is an effective way to increase the endurance capacity of older adults following a reduction in lactic acid production.” to “The current study demonstrated that supplementation of astaxanthin for 16 weeks effectively increased the endurance capacity of older adults by suppressing d-ROMs at rest and by reducing lactic acid production during the 6-min walking test. However, there were no significant intergroup differences with regard to muscle mass and strength. Therefore, the effect was most likely accompanied by an increase in endurance instead of an increase in muscle strength.”

#4. Participants were instructed not to engage in endurance activities. Was compliance with this recommendation verified?

Response: We interviewed and checked these for all subjects at the end of the experiment. Furthermore, it is necessary to notify the staff verbally or in writing when the subject goes out of this nursing home. We've checked all of these records, but we haven't been able to find any relevant subjects.

Therefore, we have added the wards for “and complied” according to the referee’s advice.

Page 2, lines 83-85, from “They were instructed to avoid any intense or unaccustomed strength training or endurance training throughout the study period.” to “They were instructed to avoid any intense or unaccustomed strength training or endurance training throughout the study period and complied.”

Reviewer 2 Report

Review on Manuscript Number: ijerph-1963164

Impacts of walking capacity on astaxanthin supplementation 2 via decreasing oxidative stress in nursing home residents

1.    Summary

The manuscript aims to investigate the effect of ingesting astaxanthin supplements on walking endurance and muscle strength in nursing home residents.

The study was conducted as healthy elderly people aged 67 to 94 were divided into two groups: twelve subjects ingesting a 24 mg/day astaxanthin supplement for 16 weeks (astaxanthin group) and eleven subjects given a placebo (group placebo). The groups were compared by measuring body components, serum d-ROMs level, 6-minute walking test and muscle strength test.

Based on the obtained results, the present study demonstrates that astaxanthin supplement is effective for preventing reduced walking endurance followed by a decline in oxidative stress in elder individuals.

In the first partIntroduction”, the authors elucidate the role of oxidative stress on the age-related decline in endurance, which is a result of the disturbance of energy metabolism and muscle strength. The importance of continuous, low-intensity exercise in reducing stress, as well as the increased risk of heart attacks and musculoskeletal injuries in the elderly, has also been clarified. Due to the reduced opportunities in these people to perform exercises, according to the study, another method should be developed to replace the exercises and act as a prevention against the increase in oxidative stress. The influence of astaxanthin on pathological processes at the cellular level has also been explained. Other studies investigating the effects of this antioxidant on the body of the elderly along with exercise, as well as possible effects on the processes without exercise, have been cited.

The Material and Method section details the voluntary enrollment, selection, and randomization of healthy elderly residents living in nursing homes to the placebo group (n = 11) or the astaxanthin group (n = 13). Also indicated are the instructions given on how to use the supplements properly, to avoid any intense or unusual strength or endurance training throughout the study period and 24 hours before the measurement sessions, and not to consume caffeinated beverages on the day of testing.

The manner in which subjects were tested before and after astaxanthin supplementation was described, and the type of measurements, namely d-ROMs test, 6-minute walk test (6 MWT), and muscle strength test, were indicated.

In the “Results” section, the results obtained from the measurements for the body weight, percent muscle mass, percent body fat, d-ROMs test, 6-minute walk test (6 MWT), and muscle strength test is presented and compared.

In the Discussion section, after analyzing the test data, conclusions are drawn that are supported by data from other authors' research.

After conducting the study, it was found that the intake of supplements containing astaxanthin in the elderly leads to an increase in energy metabolism, accompanied by the activation of ATP production, a decrease in the production of lactic acid during exercise and an accelerated utilization of lipids. As a result of all this, the endurance of the elderly is increased without increasing muscle strength.

2.    Overall opinion

The manuscript is of potential interest to researchers, pharmacists and decision makers, but needs some revision before publication to ensure better structure and flow. The text should be revised and organized.

The English format needs reworking in terms of verbal forms and correction of grammatical and typographical errors in the text.

3.    Comments

The abstract is too short. The way in which astaxanthin affects energy metabolism in the elderly, leading to increased endurance, is not explained.

What processes at the cellular level are affected by this antioxidant besides the formation of oxygen radicals and the inhibition of lipid peroxidation? Information about this is contained in lines 45 to 48, but it is good to be expanded.

Why is the number of individuals in the two groups - the placebo group (n = 11) and the astaxanthin group (n = 13) were not evened out after the selection from the originally formed groups based on diseases and physical exertion?

Correct the sentence on line 34 lexically by replacing "in" with another preposition.

The sentence on line 39 should be corrected lexically by replacing the expression "due to barriers and motivations" with another expression with a similar meaning.

On line 65, replace "through" with another preposition.

On line 66, replace the expression "a load on the body" with another, close in meaning.

The name "Table 1" on line 168 needs to be bolded.

What does the "e" mean in lines 347 and 368 before the number indicating the page the post is on?

On line 380, the journal issue number and the pages on which the publication is located are not indicated.

The conclusion is short and does not contain information about the influence of astaxanthin on the measured indicators in the elderly, which led to an increase in endurance. It is necessary to expand them.

Author Response

Response to the comments of Referee #2.

Comments:

#1. The manuscript is of potential interest to researchers, pharmacists and decision makers, but needs some revision before publication to ensure better structure and flow. The text should be revised and organized.

The English format needs reworking in terms of verbal forms and correction of grammatical and typographical errors in the text.

Response: We thank the Referee for the insightful comments on our paper. The comments have helped us to significantly improve the manuscript.

English format has been checked by a native checker in the Enago service. In addition, the modified sentence has also been checked by this service. We have reviewed the title, abstract, and text throughout.

Title: from “Impacts of walking capacity on astaxanthin supplementation via decreasing oxidative stress in nursing home residents” to “Impacts of astaxanthin supplementation on walking capacity by reducing oxidative stress in nursing home residents”

#2. The abstract is too short. The way in which astaxanthin affects energy metabolism in the elderly, leading to increased endurance, is not explained.

Response: We have corrected the abstract according to the referee’s advice.

Page 1, lines 13-30, from “Age-related decline in endurance and muscle strength is associated with oxidative stress. Astaxanthin supplement has an excellent antioxidant activity without processing any pro-oxidative properties. This study was performed to investigate the effect of ingesting astaxanthin supplements on walking endurance and muscle strength in nursing home residents. Healthy elders, aged 67 to 94 years, were divided into two groups: twelve subjects ingesting 24 mg /day astaxanthin supplement for 16 weeks (astaxanthin group), and eleven subjects given a placebo (placebo group). These subjects were compared by body component measuring, the d-ROMs level in serum, 6-min walking and muscle strength. The level of d-ROMs in the astaxanthin group significantly decreased than that in the placebo group after supplementation (p < 0.05). In addition, the distance of 6-min walking in the astaxanthin group significantly increased than that in the placebo group (p < 0.05). However, there were no significant intergroup differences in muscle mass and strength. The current study demonstrates that astaxanthin supplement is effective for preventing reduced walking endurance followed by a decline in oxidative stress in elder individuals.” to “Oxidative stress is associated with deterioration of endurance and muscle strength, which are mostly accompanied by aging. Astaxanthin supplement has excellent antioxidant activity without any pro-oxidative properties. In this study, we investigated how astaxanthin supplementation af-fects walking endurance and muscle strength in nursing home residents. Healthy elderly indi-viduals (age: 67 to 94) were divided into two groups: 13 subjects received a daily dose of 24 mg of astaxanthin for 16 weeks (astaxanthin group) and 11 subjects received a placebo (placebo group). These subjects were compared using body component measurements, serum d-ROM levels, the distance of 6-min walking, blood lactate levels after the 6-min walking test, and muscle strength. After supplementation, the levels of d-ROMs and blood lactate after the 6-min walking test in the astaxanthin group significantly decreased compared with the placebo group (p < 0.05). Additionally, the walking distance was significantly higher in the astaxanthin group than in the placebo group (p < 0.05), despite a significant reduction in lactate levels after 6-MWT (p < 0.05). However, no significant intergroup differences were observed in muscle mass and strength. Astaxanthin supplement for 16 weeks is effective to increase the endurance capacity of the elderly. Astaxanthin supplement suppresses d-ROMs at rest and lactic acid production after the 6-minute walk test. In contrast, astaxanthin supplement did not show significant intergroup differences in the muscle mass and strength. Therefore, the effect was most likely accompanied by an increase in endurance instead of an increase in muscle strength.

#3. What processes at the cellular level are affected by this antioxidant besides the formation of oxygen radicals and the inhibition of lipid peroxidation? Information about this is contained in lines 45 to 48, but it is good to be expanded.

Response: We have added the sentence about Astaxanthin effects on a cellular level and related references in the Introduction of the revised manuscript as follows:

Page 3, lines 51-55. from “Astaxanthin supplement is a red carotenoid that has excellent antioxidant activity that quenches singlet oxygen and inhibits lipid peroxidation, and it is effective as an anti-oxidant without processing any pro-oxidative properties” to “Astaxanthin supplement is a red carotenoid that provides various benefits to prevent cancer by anti-proliferation, preventing muscle atrophy by anti-apoptosis, and anti- in-flammation [9-11]. Furthermore, Astaxanthin supplement has excellent antioxidant ac-tivity that quenches singlet oxygen and inhibits lipid peroxidation, and it is effective as an anti-oxidant without processing any pro-oxidative properties.”

  1. Kurashige, M.; Okimasu, E.; Inoue, M.; Utsumi, K. Inhibition of oxidative injury of biological membranes by astaxanthin. Physiological chemistry and physics and medical NMR 1990, 22, 27-38, doi:10.1016/S0167-7799(03)00078-7.
  2. Yoshihara, T.; Yamamoto, Y.; Shibaguchi, T.; Miyaji, N.; Kakigi, R.; Naito, H.; Goto, K.; Ohmori, D.; Yoshioka, T.; Sugiura, T. Dietary astaxanthin supplementation attenuates disuse-induced muscle atrophy and myonuclear apoptosis in the rat soleus muscle. The journal of physiological sciences : JPS 2016, doi:10.1007/s12576-016-0453-4

#4. Why is the number of individuals in the two groups - the placebo group (n = 11) and the astaxanthin group (n = 13) were not evened out after the selection from the originally formed groups based on diseases and physical exertion?

Response: There were fifteen people in both groups at the start of the study. However, it was not possible to average the number of people because they dropped out in the middle of the experiment period. It was a confusing expression, so we have changed it as follows.

Page 3, Table 1. from “Did not receive allocated intervention” to “Dropped out in the middle of the intervention period”

#5. Correct the sentence on line 34 lexically by replacing "in" with another preposition.

Response: We have deleted this preposition and corrected the sentence according to the referee’s advice.

Page 1, lines 38-39, from “As a result, age-related oxidative stress conditions may contribute to a decrease in endurance by interfering with energy metabolism and muscular strength.” to “As a result, age-related oxidative stress conditions may decrease endurance by interfering with energy metabolism and muscular strength.”

#6. The sentence on line 39 should be corrected lexically by replacing the expression "due to barriers and motivations" with another expression with a similar meaning.

Response: We have deleted these words and corrected the sentence according to the referee’s advice.

Pages 1-2, lines 42-44, from “Actually, a previous study revealed that due to barriers and motivations, almost all older adults were unable to maintain exercise” to “A previous study revealed that almost all older adults were unable to continue their exercise because of constraints, such as health problems, pain, and physical environment.”

#7. On line 65, replace "through" with another preposition.

Response: We have deleted this word and corrected the sentence according to the referee’s advice.

Page 2, lines 71-72, from “This is the first study to show how astaxanthin supplements affect endurance in the elderly whose diets are well monitored through oxidative stress. to “This is the first study to show how astaxanthin supplements affect endurance in the elderly whose diets are well monitored by regulating oxidative stress levels.”

#8. On line 66, replace the expression "a load on the body" with another, close in meaning.

Response: We have deleted these words and corrected the sentence according to the referee’s advice.

Pages 2, lines 72-74, from “The unique aspect of astaxanthin supplementation is that subjects could improve endurance without a load on the body.” to “The unique aspect of astaxanthin supplementation is that the subjects could improve endurance without mechanical stress to the body.”

#9. The name "Table 1" on line 168 needs to be bolded.

Response: We have corrected the name “Table 1” according to the referee’s advice.

Page 5, Line 181. “Table 1. The body weight, percent muscle mass, and percent body fat”

#10. What does the "e" mean in lines 347 and 368 before the number indicating the page the post is on?

Response: "e" means the number of pages of papers published only electronically. We referred to previous research in this journal due to the author's guideline did not provide details on how to write the manuscript.

https://www.mdpi.com/1660-4601/19/1/449/htm

https://www.mdpi.com/1660-4601/19/1/445/htm

#11. On line 380, the journal issue number and the pages on which the publication is located are not indicated.

Response: We have corrected the reference title according to the referee’s advice.

Page 11, lines 407- 408, from “Glancy, B.; Kane, D.A.; Kavazis, A.N.; Goodwin, M.L.; Willis, W.T.; Gladden, L.B. Mitochondrial lactate metabolism: history and implications for exercise and disease. The Journal of physiology 2020.” to “Glancy, B.; Kane, D.A.; Kavazis, A.N.; Goodwin, M.L.; Willis, W.T.; Gladden, L.B. Mitochondrial lactate metabolism: history and implications for exercise and disease. The Journal of physiology 2020. 99, 863-888, doi:10.1113/JP278930.”

In addition, we have added the doi for all references.

#12. The conclusion is short and does not contain information about the influence of astaxanthin on the measured indicators in the elderly, which led to an increase in endurance. It is necessary to expand them.

Response: We have corrected the conclusion according to the referee’s advice.

Page 9, lines 309-314, from “The current study demonstrates that suppressing oxidative stress by taking astaxanthin supplement for 16 weeks is an effective way to increase the endurance capacity of older adults following a reduction in lactic acid production.” to “The current study demonstrated that supplementation of astaxanthin for 16 weeks effectively increased the endurance capacity of older adults by suppressing d-ROMs at rest and by reducing lactic acid production during the 6-min walking test. However, there were no significant intergroup differences with regard to muscle mass and strength. Therefore, the effect was most likely accompanied by an increase in endurance instead of an increase in muscle strength.”

Round 2

Reviewer 1 Report

Thank you for explaining and answering your questions. I recommend making some minor adjustments:

The reasons why the participants were given the same amount of astaxanthin supplement (not the amount graded by weight) I recommend to state within the limits of the work or explain in the methods section.

Consider including mention of a possible height measurement error in the text.

Author Response

Victoria Liu, PhD

Section Managing Editor,International Journal of Environmental Research and Public Health

Dear Dr. Liu:

We are submitting a revised version of our manuscript (ijerph-1963164) entitled “Impacts of astaxanthin supplementation on walking capacity by reducing oxidative stress in nursing home residents” by Nakanishi et al. We have addressed the additional comments of Referee #1 on a point-by-point basis below. We appreciate the time and effort that the Senior Editor and Referees have taken to make helpful comments and we believe that the manuscript has been improved based on these comments. 

We hope that our responses are satisfactory to make the manuscript suitable for publication in International Journal of Environmental Research and Public Health as a research article.

Sincerely yours,

Hidemi Fujino, Ph.D.

Professor

Kobe University Graduate School of Health Sciences

Department of Rehabilitation Science

7-10-2 Tomogaoka, Suma-ku, Kobe 654-0142, Japan

Telephone: +81-78-796-4582

Fax: +81-78-796-4582

Response to the comments of Referee #1,

We thank the Referee for carefully reading our manuscript and for the helpful comments.

Comments:

#1. The reasons why the participants were given the same amount of astaxanthin supplement (not the amount graded by weight) I recommend to state within the limits of the work or explain in the methods section.

Response: We have added the limitation in discussion according to the referee’s advice.

Page 9, lines 307-313, “Another limitation is that we only studied the effects of the same amount of astaxanthin supplement; however, we did not attempt to elucidate the effects of different weights of astaxanthin. We used soft capsules for our experiments to hide the appearance of astaxanthin color. Therefore, all subjects received the same amount of astaxanthin sup-plement owing to the difficulty of changing the content of each capsule. Future studies should determine the effects of the amount graded by weight on these results.”

#2. Consider including mention of a possible height measurement error in the text.

Response: We have added the limitation in discussion according to the referee’s advice.

Page 2, lines 313-317, “In this study, a height measurement error was observed. Correcting the alignment is necessary to measure the height of the elderly; furthermore, it is speculated that this measurement error may have occurred as a result of the degree of correction. Minimizing measurement errors in elderly research is of paramount importance in spite of performing alignment corrections within a range that does not cause discomfort to the subjects.”
